# Cite-seeing and reviewing: A study on citation bias in peer review

**Ivan Stelmakh**[1,2☯]*, **Charvi Rastogi**[3☯], **Ryan Liu**[3], **Shuchi Chawla**[4], **Federico Echenique**[5], **Nihar B. Shah**[3]

**1** New Economic School, Moscow, Russia, **2** Yakov & Partners, Moscow, Russia, **3** School of Computer Science, Carnegie Mellon University, Pittsburgh, Pennsylvania, United States of America, **4** Department of Computer Science, University of Texas at Austin, Austin, Texas, United States of America, **5** Division of the Humanities and Social Sciences, California Institute of Technology, Pasadena, California, United States of America

☯ These authors contributed equally to this work.
* ivan_stelmakh@yakov.partners

## Abstract

Citations play an important role in researchers' careers as a key factor in evaluation of scientific impact. Many anecdotes advice authors to exploit this fact and cite prospective reviewers to try obtaining a more positive evaluation for their submission. In this work, we investigate if such a *citation bias* actually exists: Does the citation of a reviewer's own work in a submission cause them to be positively biased towards the submission? In conjunction with the review process of two flagship conferences in machine learning and algorithmic economics, we execute an observational study to test for citation bias in peer review. In our analysis, we carefully account for various confounding factors such as paper quality and reviewer expertise, and apply different modeling techniques to alleviate concerns regarding the model mismatch. Overall, our analysis involves 1,314 papers and 1,717 reviewers and detects citation bias in both venues we consider. In terms of the effect size, by citing a reviewer's work, a submission has a non-trivial chance of getting a higher score from the reviewer: an expected increase in the score is approximately 0.23 on a 5-point Likert item. For reference, a one-point increase of a score by a single reviewer improves the position of a submission by 11% on average.

## 1 Introduction

Peer review is the backbone of academia. Across many fields of science, peer review is used to decide on the outcome of manuscripts submitted for publication. Moreover, funding bodies in different countries employ peer review to distribute multi-billion dollar budgets through grants and awards. Given that stakes in peer review are high, it is extremely important to ensure that evaluations made in the review process are not biased by factors extraneous to the submission quality. This requirement is especially important in presence of the Matthew effect ("rich get richer") in academia [1]: an advantage a researcher receives by publishing even a single work in a prestigious venue or getting a research grant early may have far-reaching consequences on their career trajectory.

**Data Availability Statement:** We note that the release of experimental data would compromise the reviewers' confidentiality. Thus, following prior works that empirically analyze the conference peer-review process (Tomkins et al., 2017; Shah et al.,

2018; Lawrence and Cortes, 2014), and complying with the conference's policy, we are unable to release the data and code from the experiment.

**Funding:** NSF CAREER award 1942124 was awarded to Nihar Shah (https://www.nsf.gov/awardsearch/showAward?AWD_ID=1942124&HistoricalAwards=false) J.P. Morgan AI research fellowship was awarded to Charvi Rastogi (https://www.jpmorgan.com/technology/artificial-intelligence/research-awards/phd-fellowship-2021) The funders had no role in study design, data collection and analysis, decision to publish, or preparation of the manuscript. There was no additional external funding received for this study.

**Competing interests:** The authors have declared that no competing interests exist.

The key decision-makers in peer review are fellow researchers with expertise in the research areas of the submissions they review. While there is no doubt that the primary incentive for reviewers is the opportunity to contribute to the development of the field, a structure of the performance indicators in academia may create additional incentives that would require a rational reviewer to deviate from truthful behavior. Specifically, a metric on which researchers' success is often gauged is the number of citations they receive. A Google Scholar profile, for example, summarizes a researcher's output in the total number of citations to their work, and several other citation-based metrics (h-index, i10-index). Citations are also considered to be an important factor in tenure and promotion decisions [2, 3]. This raises the possibility that reviewers may, consciously or subconsciously, be more lenient towards submissions that cite their work as acceptance of this submission is beneficial for the reviewer's own career success.

Several cases document extreme manifestations of such side incentives. The Committee on Publication Ethics [4] reports that a handling editor of an unnamed journal asked authors to add citations to their work more than 50 times, three times more often than they asked authors to add citations of papers they did not co-author. The editorial team of the journal did not find a convincing scientific justification for such requests and the handling editor resigned from their duties. A similar case [5] was uncovered in the Journal of Theoretical Biology where an editor was asking authors to add 35 citations on average to each submitted paper, and 90% of these requests were to cite papers authored by that editor. This behavior of the editor traced back to decades before being uncovered, and furthermore, authors had complied to such requests with an "apparently amazing frequency".

While the aforementioned studies report some cases of gross scientific misconduct, they do not quantify the impact of citations on the dynamics of reviewing-as-usual. Of course, most scientists do not engage in forcing the authors to add unnecessary citations to their past work. However, consciously or subconsciously, they may get positively biased when they notice that a paper they review cites some of their past work. Thus, in the present work, *we investigate whether reviewers are biased by citations*.

Note that citation of a reviewer's past work may impact the reviewer's evaluation of a submission in two ways: first, it can impact the scientific merit of the submission, thereby causing a *genuine change* in evaluation; second, it can induce an *undesirable bias* in evaluation that goes beyond the genuine change. We use the term "*citation bias*" to refer to the second mechanism. Formally, the research question we investigate in this work is as follows:

**Research question**: Does the citation of a reviewer's work in a submission *cause* the reviewer to be positively *biased* towards the submission, that is, *cause* a shift in reviewer's evaluation that goes beyond the genuine change in the submission's scientific merit?

Citation bias, if present, contributes to the unfairness of academia by making peer-review decisions dependent on factors irrelevant to the submission quality. It is therefore important for stakeholders to understand if citation bias is present, and whether it has a strong impact on the peer-review process.

Two studies have previously investigated citation bias in peer review [6, 7]. These studies analyze journal and conference review data and report mixed evidence of citation bias in reviewers' recommendations. However, their analysis does not account for confounding factors such as paper quality (stronger papers may have longer bibliographies) or reviewer expertise (cited reviewers may have higher expertise). Thus, past works do not decisively answer the question of the presence of citation bias. A more detailed discussion of these and other relevant works is provided in Section 2.

The existence of citation bias can also skew incentives for authors. In fact, the speculation of existence of citation bias has resulted in anecdotal comments of the form [8]: *"We all know of cases where including citations to journal editors or potential reviewers [. . .] will help a paper's chances of being accepted for publication in a specific journal."* Furthermore, existing research documents that the suggestion to pad reference lists with unnecessary citations is taken seriously by some authors. For example, a survey conducted by [9] indicates that over 40% of authors across several disciplines would preemptively add non-critical citations to their journal submission when the journal has a reputation of asking for such citations. The same observation applies to grant proposals, with 15% of authors willing to add citations even when "*those citations are of marginal import to their proposal*". Thus anecdotes and surveys suggest that many authors perceive the citation bias to be present in reviewing.

With these motivations, in this work we conduct a quantitative investigation to (un)substantiate this intuition. If citation bias exists, then peer-reviewing systems should take measures to counteract it. On the other hand, if citation bias does not exist, then it presents some reassurance in terms of the objectivity of peer review, and also informs authors about the futility of the practice of padding citations.

## 1.1 Our contributions

In this work, we investigate the research question in a large-scale study conducted in conjunction with the review process of two flagship publication venues: 2020 International Conference on Machine Learning (ICML 2020) and 2021 ACM Conference on Economics and Computation (EC 2021). We execute a carefully designed observational analysis that accounts for various confounding factors such as paper quality and reviewer expertise. Overall, our analysis identifies citation bias in both venues we consider: by adding a citation of a reviewer, a submission can increase the expectation of the score given by the reviewer by 0.23 (on a 5-point scale) in EC 2021 and by up to 0.42 (on a 6-point scale) in ICML 2020. For better interpretation of the effect size, we note that on average, a one-point increase in a score given by a single reviewer improves the position of a submission by 11%.

Finally, it is important to note that the bias we investigate is not necessarily an indicator of unethical behavior on the part of reviewers or authors. Citation bias may be present even when reviewers do not consciously attempt to champion papers that cite their past work, and when authors do not try to deliberately cite potential reviewers. Crucially, even subconscious citation bias is problematic for fairness reasons. Thus, understanding whether the bias is present is important for improving peer-review practices and policies.

## 2 Related literature

In this section, we discuss relevant past studies. We begin with an overview of cases, anecdotes, and surveys that document practices of coercive citations. We then discuss two works that perform statistical testing for citation bias in peer review. Finally, we conclude with a list of works that test for other biases in the peer-review process. We refer the reader to [10] for a broader overview of literature on peer review.

### 2.1 Coercive citations

In the introduction, we discussed several studies [4, 5] that mention the practice of coercive citations—spurious requests to add citations to the submission from editors of the journals. Another study [9] provides additional evidence of coercion. They conduct a survey which reveals that 14.1% of approximately 12,000 respondents from different research areas have experienced coercion by journal editors. [11] notes that coercion happens not only at the

journal level, but also at the level of individual reviewers. Specifically, 22.7% of 220 researchers from the National Institute of Environmental Health Sciences who participated in the survey reported that they have received reviews requesting them to include unnecessary references to publications authored by the reviewer.

Given such evidence of coercion, it is not surprising that authors are willing to preemptively inflate bibliographies of their submissions either because journals they submit to have a reputation for coercion [9] or because they hope to bias reviewers and increase the chances of the submission [12]. That said, observe that evidence we discussed above is based on either case studies or surveys of authors' perceptions. We note, however, that (i) authors in peer review usually do not know identities of reviewers, and hence may incorrectly perceive a reviewer's request to cite someone else's work as that of coercion to cite the reviewer's own work; and (ii) case studies describe only the most extreme cases and are not necessarily representative of the average practice. Thus, the aforementioned findings could overestimate the prevalence of coercion and do not necessarily imply that a submission can significantly boost its acceptance chances by strategically citing potential reviewers.

## 2.2 Citation bias

We now describe several other works that investigate the presence of citation bias in peer review. First, Sugimoto and Cronin [6] analyze the editorial data of the Journal of the American Society of Information Science and Technology and study the relationship between the reviewers' recommendations and the presence of references to reviewers' works in submissions. They find mixed evidence of citation bias: a statistically significant difference between *accept* and *reject* recommendations (cited reviewers are more likely to recommend acceptance than reviewers who are not cited) becomes insignificant if they additionally consider *minor/major revision* decisions. We note, however, that the analysis of [6] computes correlations and does not control for confounding factors associated with paper quality and reviewer identity (see discussion of potential confounding factors in Section 3.2.1). Thus, that analysis does not allow to test for the causal effect. Work by Schriger et. al. [13] follows a similar narrative, where they find a difference in the scores of cited and uncited reviewers in the review process of the Annals of Emergency Medicine. However, they conclude that the difference may be explained by cited reviewers being assigned higher quality submissions.

Another work [7] performs data analysis of the 2010 edition of ACM Internet Measurement Conference and reports findings that suggest the presence of citation bias. As a first step of the analysis, they compute a correlation between acceptance decisions and the number of references to papers authored by 2010 TPC (technical program committee) members. For long papers, the correlation is 0.21 ($n = 109$, $p < 0.03$) and for short papers the correlation is 0.15 ($n = 102$, $p = 0.12$). Similar to the analysis of [6], these correlations do not establish causal relationship due to unaccounted confounding factors such as paper quality (papers relevant to the venue may be more likely to cite members of TPC than out-of-scope papers).

To mitigate confounding factors, [7] perform a second step of the analysis. They recompute correlations but now use members of the *2009* TPC who are not in 2010 TPC as a target set of reviewers. Reviewers from this target set did not impact the decisions of the 2010 submissions and hence this second set of correlations can serve as an unbiased contrast. For long papers, the contrast correlation is 0.13 ($n = 109$, $p = 0.19$) and for short papers, the contrast correlation is $-0.04$ ($n = 102$, $p = 0.66$). While the *difference* between actual and contrast correlations hints at the presence of citation bias, we note that (i) the sample size of the study may not be sufficient to draw statistically significant conclusions (the paper does not formally test for significance of the difference); (ii) the overlap between 2010 and 2009 committees is itself a

confounding factor—members in the overlap may be statistically different (e.g., more senior) from those present in only one of the two committees.

## 2.3 Testing for other biases in peer review

A long line of literature [14–21, and many others] scrutinizes the peer-review process for various biases. These works investigate gender, fame, positive-outcome, and many other biases that can hurt the quality of the peer-review process. Our work continues this line by investigating citation bias.

## 3 Methods

In this section, we outline the design of the experiment we conduct to investigate the research question of this paper. Section 3.1 introduces the venues in which our experiment was executed and discusses details of the experimental procedure. Section 3.2 describes our approach to the data analysis. In what follows, for a given pair of submission $\mathcal{S}$ and reviewer $\mathcal{R}$, we say that reviewer $\mathcal{R}$ is CITED in $\mathcal{S}$ if one or more of their past papers are cited in the submission. Otherwise, reviewer $\mathcal{R}$ is UNCITED.

## 3.1 Experimental procedure

We begin with a discussion the details of the experiment we conduct in this work.

**3.1.1 Experimental setting.** The experiment was conducted in the peer-review process of two conferences:

- **ICML 2020** International Conference on Machine Learning is a flagship machine learning conference that receives thousands of paper submissions and manages a pool of thousands of reviewers.

- **EC 2021** ACM Conference on Economics and Computation is the top conference at the intersection of computer science and economics. The conference is smaller than ICML and handles several hundred submissions and reviewers.

It is important to note that in the field of computer science, conferences are considered to be a final publication venue for research and are typically ranked higher than journals. Full papers are reviewed in CS conferences, and their publication has archival value. Now, rows 1 and 2 of Table 1 display information about the size of the conferences used in the experiment.

The peer-review process in both venues is organized in a double-blind manner (neither authors nor reviewers know the identity of each other) and follows the conventional pipeline that we now outline. After the submission deadline, reviewers indicate their preference in reviewing the submissions. Additionally, program chairs compute measures of similarity between submissions and reviewers which are based on (i) overlap of research topics of

**Table 1. Statistics on the venues where the experiment is executed.**

|  | ICML 2020 | EC 2021 |
|---|---|---|
| # REVIEWERS | 3,064 | 154 |
| # SUBMISSIONS | 4,991 | 496 |
| NUMBER OF SUBMISSIONS WITH AT LEAST ONE CITED REVIEWER | 1,513 | 287 |
| FRACTION OF SUBMISSIONS WITH AT LEAST ONE CITED REVIEWER | 30% | 58% |

Table notes: The number of reviewers includes all regular reviewers. The number of submissions includes all submissions that were not withdrawn from the conference by the end of the initial review period.

submissions/reviewers (both conferences) and (ii) semantic overlap [22] between texts of submissions' and reviewers' past papers (ICML). All this information is then used to assign submissions to reviewers who have several weeks to independently write initial reviews. The initial reviews are then released to authors who have several days to respond to these reviews. Finally, reviewers together with more senior members of the program committee engage in the discussions and make final decisions, accepting about 20% of submissions to the conference.

**3.1.2 Intervention.**   As we do not have control over bibliographies of submissions, we cannot intervene on the citation relationship between submissions and reviewers. We rely instead on the analysis of observational data. As we explain in Section 3.2, for our analysis to have a strong detection power, it is important to assign a large number of submissions to both CITED and UNCITED reviewers. In ICML, this requirement is naturally satisfied due to its large sample size, and we assign submissions to reviewers using the PR4A assignment algorithm [23] that does not specifically account for the citation relationship in the assignment.

The number of papers submitted to the EC 2021 conference is much smaller. Thus, we tweak the assignment process in a manner that gets us a larger sample size while retaining the conventional measures of the assignment quality. To explain our intervention, we note that, conventionally, the quality of the assignment in the EC conference is defined in terms of satisfaction of reviewers' preferences in reviewing the submissions, and research topic similarity. However, in addition to being useful for the sample size of our analysis, citation relationship has also been found [24] to be a good indicator for the review quality and was used in other studies to measure similarity between submissions and reviewers [25]. With this motivation, in EC, we use an adaptation of the popular TPMS assignment algorithm [22] with the objective consisting of two parts: (i) conventional measure of the assignment quality and (ii) the number of CITED reviewers in the assignment. We then introduce a parameter that can be tuned to balance the two parts of the objective and find an assignment that has a large number of CITED reviewers while not compromising the conventional metrics of assignment quality. Additionally, the results of the automated assignment are validated by senior members of the program committee who can alter the assignment if some (submission, reviewer) pairs are found unsuitable. As a result, Table 1 demonstrates that in the final assignment more than half of the EC 2021 submissions were assigned to at least one CITED reviewer.

**3.1.3 Ethics statement.**   This study was analyzed by Carnegie Mellon University's Institutional Review Board that approved the study by deciding it to be exempt from review. We did not notify reviewers about the experiment or collect consent, since this was an observational study.

## 3.2 Analysis

As we mentioned in the previous section, in this work we rely on analysis of observational data. Specifically, our analysis operates with *initial reviews* that are written independently before author feedback and discussion stages (see description of the review process in Section 3.1). As is always the case for observational studies, our data can be affected by various confounding factors. Thus, we design our analysis procedure to alleviate the impact of several plausible confounders. In Section 3.2.1 we provide a list of relevant confounding factors that we identify and in Section 3.2.2 we explain how our analysis procedure accounts for them.

**3.2.1 Confounding factors.**   We begin by listing the confounding factors that we account for in our analysis. For ease of exposition, we provide our description in the context of a naïve approach to the analysis and illustrate how each of the confounding factors can lead to false

conclusions of this naïve analysis. The naïve analysis we consider compares the mean of numeric evaluations given by all CITED reviewers to the mean of numeric evaluations given by all UNCITED reviewers and declares bias if these means are found to be unequal for a given significance level. With these preliminaries, we now introduce the confounding factors.

1. C1 **Genuinely missing citations** Each reviewer is an expert in their own work. Hence, it is easy for reviewers to spot a genuinely missing citation to their own work, such as missing comparison to their own work that has a significant overlap with the submission. At the same time, reviewers may not be as familiar with the papers of other researchers and their evaluations may not reflect the presence of genuinely missing citations to these papers. Therefore, the scores given by UNCITED reviewers could be lower than scores of CITED reviewers even in absence of citation bias, which would result in the naïve test declaring the effect when the effect is absent.

2. C2 **Paper quality** As shown in Table 1, not all papers submitted to the EC and ICML conferences were assigned to CITED reviewers. Thus, reviews by CITED and UNCITED reviewers were written for intersecting, but not identical, sets of papers. Among papers that were not assigned to CITED reviewers there could be papers which are clearly out of the conference's scope. Thus, even in absence of citation bias, there could be a difference in evaluations of CITED and UNCITED reviewers caused by the difference in relevance between two groups of papers the corresponding reviews were written for. The naïve test, however, will raise a false alarm and declare the bias even though the bias is absent.

3. C3 **Reviewer expertise** The reviewer and submission pools of the ICML and EC conferences are diverse and submissions are assigned to reviewers of different expertise in reviewing them. The expertise of a reviewer can be simultaneously related to the citation relationship (expert reviewers may be more likely to be CITED) and to the stringency of evaluations (expert reviewers may be more lenient or strict). Thus, the naïve analysis that ignores this confounding factor is in danger of raising a false alarm or missing the effect when it is present.

4. C4 **Reviewer preference** As we mentioned in Section 3.2.2, the assignment of submissions to reviewers is, in part, based on reviewers' preferences. Thus, (dis-)satisfaction of the preference may impact reviewers' evaluations—for example, reviewers may be more lenient towards their top choice submissions than to submissions they do not want to review. Since citation relationships are not guaranteed to be independent of the reviewers' preferences, the naïve analysis can be impacted by this confounding factor.

5. C5 **Reviewer seniority** Some past work has observed that junior reviewers may sometime be stricter than their senior colleagues [26, 27, note that some other works such as [28, 29] do not observe this effect]. If senior reviewers are more likely to be CITED (e.g., because they have more papers published) and simultaneously are more lenient, the seniority-related confounding factor can bias the naïve analysis.

**3.2.2 Analysis procedure.** Having introduced the confounding factors, we now discuss the analysis procedure that alleviates the impact of these confounding factors and enables us to investigate the research question. Specifically, our analysis consists of two steps: data filtering and inference. For ease of exposition, we first describe the inference step and then the filtering step.

**3.2.2.1 Inference**. The key quantities of our inference procedure are overall scores (score) given in initial reviews and binary indicators of the citation relationship

(`citation`). Overall scores represent recommendations given by reviewers and play a key role in the decision-making process. Thus, a causal connection between `citation` and `score` is a strong indicator of citation bias in peer review.

To test for causality, our inference procedure accounts for confounders C2–C5 (confounder C1 is accounted for in the filtering step). To account for these confounders, for each (submission, reviewer) pair we introduce several characteristics which we now describe, ignoring non-critical differences between EC and ICML. S1 Appendix provides more details on how these characteristics are defined in the two individual venues.

- `quality` Relevant quality of the submission for the publication venue considered. We note that this quantity can be different from the quality of the submission independent of the publication venue. The value of relative quality of a submission is, of course, unknown and below we explain how we accommodate this variable in our analysis to account for confounder C2.

- `expertise` Measure of expertise of the reviewer in reviewing the submission. In both ICML and EC, reviewers were asked to self-evaluate their ex post expertise in reviewing the assigned submissions. In ICML, two additional expertise-related measures were obtained: (i) ex post self-evaluation of the reviewer's confidence; (ii) an overlap between the text of each submitted paper and each reviewer's past papers [22]. We use all these variables to control for confounding factor C3.

- `preference` Preference of the reviewer in reviewing the submission. As we mentioned in Section 3.1, both ICML and EC conferences elicited reviewers' preferences in reviewing the submissions. We use these quantities to alleviate confounder C4.

- `seniority` An indicator of reviewers' seniority. For the purpose of decision-making, both conferences categorized reviewers into two groups. While specific categorization criteria were different across conferences, conceptually, groups were chosen such that one contained more senior reviewers than the other. We use this categorization to account for the seniority confounding factor C5.

Having introduced the characteristics we use to control for confounding factors C2–C5, we now discuss the two approaches we take in our analysis.

*3.2.2.1.1 Parametric approach (EC and ICML).* First, following past observational studies of the peer-review procedure [17, 30] we assume a linear approximation of the `score` given by a reviewer to a submission:

$$\texttt{score} \sim \alpha_0 \quad +\alpha_1 \cdot \texttt{quality} + \alpha_2 \cdot \texttt{expertise} + \alpha_3 \cdot \texttt{preference}$$
$$+\alpha_4 \cdot \texttt{seniority} + \alpha^* \cdot \texttt{citation}. \tag{1}$$

Here, the notation $y \sim \alpha_0 + \sum_i^n \alpha_i x_i$ means that given values of $\{x_i\}_{i=1}^n$, dependent variable $y$ is distributed as a Gaussian random variable with mean $\alpha_0 + \sum_i^n \alpha_i x_i$ and variance $\sigma^2$. The values of $\{\alpha_i\}_{i=0}^n$ and $\sigma$ are unknown and need to be estimated from data. Variance $\sigma^2$ is independent of $\{x_i\}_{i=1}^n$. Under this assumption, the test for citation bias as formulated in our research question reduces to the test for significance of $\alpha^*$ coefficient. However, we cannot directly fit the data we have into the model as the values of `quality` are not readily available. Past work [17] uses a heuristic to estimate the values of paper quality, however, this approach was demonstrated [31] to be unable to reliably control the false alarm probability.

To avoid the necessity to estimate `quality`, we restrict the set of papers used in the analysis to papers that were assigned to at least one CITED reviewer and at least one UNCITED reviewer.

At the cost of the reduction of the sample size, we are now able to take a difference between scores given by CITED and UNCITED reviewers *to the same submission* and eliminate `quality` from the model (1). As a result, we apply a standard tools for the linear regression inference to test for the significance of the target coefficient $\alpha^*$. We refer the reader to S2 Appendix for more details on the parametric approach.

***3.2.2.1.2 Non-parametric approach (ICML)*.** While the parametric approach we introduced above is conventionally used in observational studies of peer review and offers strong detection power even for small sample sizes, it relies on strong modeling assumptions that are not guaranteed to hold in the peer-review setting [31]. To overcome these limitations, we also execute an alternative non-parametric analysis that we now introduce.

The idea of the non-parametric analysis is to match (submission, reviewer) pairs on the values of all four characteristics (`quality`, `expertise`, `preference`, and `seniority`) while requiring that matched pairs have different values of `citation`. As in the parametric analysis, we overcome the absence of access to the values of `quality` by matching (submission, reviewer) pairs *within* each submission. In this way, we ensure that matched (submission, reviewer) pairs have the same values of confounding factors C2–C5. We then compare mean scores given by CITED and UNCITED reviewers, focusing on the restricted set of matched (submission, reviewer) pairs, and declare the presence of citation bias if the difference is statistically significant. Again, more details on the non-parametric analysis are given in S3 Appendix.

**3.2.2.1 Data filtering**. The purpose of the data-filtering procedure is twofold: first, we deal with missing values; second, we take steps to alleviate the genuinely missing citations confounding factor C1.

***3.2.2.1.1 Missing values*.** As mentioned above, for a submission to qualify for our analysis, it should be assigned to at least one CITED reviewer and at least one UNCITED reviewer. In ICML data, 578 out of 3,335 (submission, reviewer) pairs that qualify for the analysis have values of certain variables corresponding to `expertise` and `preference` missing. The missingness of these values is due to various technicalities: reviewers not having profiles in the system used to compute textual overlap or not reporting preferences in reviewing submissions. Thus, given a large size of the ICML data, we remove such (submission, reviewer) pairs from the analysis.

In the EC conference, the only source of missing data is reviewers not entering their `preference` in reviewing some submissions. Out of 849 (submission, reviewer) pairs that qualify for the analysis, 154 have reviewer's `preference` missing. Due to a limited sample size, we do not remove such (submission, reviewer) pairs from the analysis and instead accommodate missing preferences in our parametric model (1) (see S1 and S2 Appendices for details).

***3.2.2.1.2 Genuinely missing citation*.** Another purpose of the filtering procedure is to account for the genuinely missing citations confounder C1. The idea of this confounder is that even in absence of citation bias, reviewers may legitimately decrease the score of a submission because citations to some of their own past papers are missing. The frequency of such legitimate decreases in scores may be different between CITED and UNCITED reviewers, resulting in a confounding factor. To alleviate this issue, we aim at identifying submissions with genuinely missing citations of reviewers' past papers and removing them from the analysis. More formally, to account for confounder C1, we introduce the following exclusion criteria:

**Exclusion criteria**: The reviewer flags a missing citation of *their own* work and this complaint is valid for reducing the score of the submission

The specific implementation of a procedure to identify submissions satisfying this criteria is different between ICML and EC conferences and we introduce it separately.

*EC*. In the EC conference, we added a question to the reviewer form that asked reviewers to report if a submission has some important relevant work missing from the bibliography. Among 849 (submission, reviewer) pairs that qualify for inclusion to our inference procedure, 110 had a corresponding flag raised in the review. For these 110 pairs, authors of the present paper (CR, FE) manually analyzed the submissions and the reviews, identifying submissions that satisfy the exclusion criteria. CR conducted an initial, basic screening and all cases that required a judgement were resolved by FE—a program chair of the EC 2021 conference.

Overall, among the 110 target pairs, only three requests to add citations were found to satisfy the exclusion criteria. All (submission, reviewer) pairs for these three submissions were removed from the analysis, ensuring that reviews written in the remaining (submission, reviewer) pairs are not susceptible to confounding factor C1.

*ICML*. In ICML, the reviewer form did not have a flag for missing citations. Hence, to fully alleviate the genuinely missing citations confounding factor, we would need to analyze all the 1,617 (submission, UNCITED reviewer) pairs qualifying for the inference step to identify those satisfying the aforementioned exclusion criteria. Note that, in principle, CITED reviewers may also legitimately decrease the score because the submission misses some of their past papers. However, this reduction in score would lead us to an underestimation of the effect (or, under the absence of citation bias, to the counterintuitive direction of the effect) and hence we tolerate it.

We begin from the analysis of (submission, UNCITED reviewer) pairs that qualify for our non-parametric analysis. There are 63 such pairs and analysis conducted by an author of the present paper (IS—a workflow chair of ICML 2020) found that three of them satisfy the exclusion criteria. The corresponding three submissions were removed from our non-parametric analysis.

The fraction of (submission, UNCITED reviewer) pairs with a genuinely missing citation of the reviewer's past paper in ICML is estimated to be 5% ($^3/_{63}$). As this number is relatively small, the impact of this confounding factor is limited. In absence of the missing citation flag in the reviewer form, we decided not to account for this confounding factor in the parametric analysis of the ICML data. Thus, we urge the reader to be aware of this confounding factor when interpreting the results of the parametric inference.

## 4 Results

As described in Section 3, we study our research question using data from two venues (ICML 2020 and EC 2021) and applying two types of analysis (parametric for both venues and non-parametric for ICML). While the analysis is conducted on observational data, we intervene in the assignment stage of the EC conference in order to increase the sample size of our study. Table 2 displays the key details of our analysis (first group of rows) and numbers of unique submissions, reviewers, and (submission, reviewer) pairs involved in our analysis (second group of rows).

The dependent variable in our analysis is the score given by a reviewer to a submission in the initial independent review. Therefore, the key quantity of our analysis (test statistic) is an expected increase in the reviewer's score due to citation bias. In EC, reviewers scored submissions on a 5-point Likert item while in ICML a 6-point Likert item was used. Thus, the test statistic can take values from -4 to 4 in EC and from -5 to 5 in ICML. Positive values of the test statistic indicate the positive direction of the bias and the absolute value of the test statistic captures the magnitude of the effect.

The third group of rows in Table 2 summarizes the key results of our study. Overall, we observe that after accounting for confounding factors, all three analyses detect statistically

**Table 2. Results of the analysis.**

| | | EC 2021 | ICML 2020 | ICML 2020 |
|---|---|---|---|---|
| ANALYSIS | | PARAMETRIC | PARAMETRIC | NON-PARAMETRIC |
| INTERVENTION | | ASSIGNMENT STAGE | NO | NO |
| MISSING VALUES | | INCORPORATED | REMOVED | REMOVED |
| GENUINELY MISSING CITATIONS | | REMOVED | UNACCOUNTED ($\sim$5%) | REMOVED |
| SAMPLE SIZE | # SUBMISSIONS (S) | 283 | 1,031 | 60 |
| | # REVIEWERS (R) | 152 | 1,565 | 115 |
| | # (S, R)-PAIRS | 840 | 2,757 | 120 |
| TEST STATISTIC | | 0.23 ON 5-POINT SCALE | 0.16 ON 6-POINT SCALE | 0.42 ON 6-POINT SCALE |
| TEST STATISTIC (95% CI) | | [0.06, 0.40] | [0.05, 0.27] | [0.10, 0.73] |
| P VALUE | | 0.009 | 0.004 | 0.02 |

Table notes: The results suggest that citation bias is present in both EC 2021 and ICML 2020 conferences. *P* values and confidence intervals for parametric analysis are computed under the standard assumptions of linear regression. For non-parametric analysis, *P* value is computed using permutation test and the confidence interval is bootstrapped. All *P* values are two-sided.

significant differences between the behavior of CITED and UNCITED reviewers (see the last row of the table for *P* values). Thus, we conclude that citation bias is present in both ICML 2020 and EC 2021 venues.

We note that conclusions of the parametric analysis are contingent upon satisfaction of the linear model assumptions and it is a priori unclear if these assumptions are satisfied to a reasonable extent. To investigate potential violation of these assumptions, in S4 Appendix we conduct analysis of model residuals. This analysis suggests that linear models provide a reasonable fit to both ICML and EC data, thereby supporting the conclusions we make in the main analysis. Additionally, we note that our non-parametric analysis makes less restrictive assumptions on reviewers' decision-making but still arrives at the same conclusion.

## 4.1 Effect size

To interpret the effect size, we note that the value of the test statistic captures the magnitude of the effect. In EC 2021, a citation of reviewer's paper would result in an expected increase of 0.23 in the score given by the reviewer. Similarly, in ICML 2020 the corresponding increase would be 0.16 according to the parametric analysis and 0.42 according to the non-parametric analysis. Confidence intervals for all three point estimates (rescaled to 5-point scale) overlap, suggesting that the magnitude of the effect is similar in both conferences. Overall, the values of the test statistic demonstrate that a citation of a reviewer results in a considerable improvement in the expected score given by the reviewer. In other words, there is a non-trivial probability of reviewer increasing their score by one or more points when cited. With this motivation, to provide another interpretation of the effect size, we now estimate the effect of a one-point increase in a score by a single reviewer on the outcome of the submission.

Specifically, we first rank all submissions by the mean score given in the initial reviews, breaking ties uniformly at random. For each submission, we then compute the improvement of its position in the ranking if one of the reviewers increases their score by one point. Finally, we compute the mean improvement over all submissions to arrive at the average improvement. As a result, on average, in both conferences a one-point increase in a score given by a single reviewer improves the position of a submission in a score-based ordering by 11%. Thus, having a reviewer who is cited in a submission can have a non-trivial implication on the acceptance chances of the submission.

As a note of caution, in actual conferences decisions are based not only on scores, but also on the textual content of reviews, author feedback, discussions between reviewers, and other factors. We use the readily available score-based measure to obtain a rough interpretation of the effect size, but we encourage the reader to keep these qualifications in mind when interpreting the result.

## 5 Discussion

We have reported the results of two observational studies of citation bias conducted in flagship machine learning (ICML 2020) and algorithmic economics (EC 2021) conferences. To test for the causal effect, we carefully account for various confounding factors and rely on two different analysis approaches. Overall, the results suggest that citation bias is present in peer-review processes of both venues. A considerable effect size of citation bias can (i) create a strong incentive for authors to add superfluous citations of potential reviewers, and (ii) result in unfairness of final decisions. Thus, the finding of this work may be informative for conference chairs and journal editors who may need to develop measures to counteract citation bias in peer review. In this section, we provide additional discussion of several aspects of our work.

### 5.1 Observational caveat

First, we want to underscore that, while we try to carefully account for various confounding factors and our analysis employs different techniques, our study remains observational. Thus, the usual caveat of unaccounted confounding factors applies to our work. The main assumption that we implicitly make in this work is that the list of confounding factors C1–C5 is (i) exclusive and (ii) can be adequately modelled with the variables we have access to. As an example of a violation of these assumptions, consider that CITED reviewers could possess some characteristic that is not captured by `expertise`, `preference`, and `seniority` and makes them more lenient towards the submission they review. In this case, the effect we find in this work would not be a causation. That said, we note that to account for confounding factors, we used all the information that is routinely used in many publication venues to describe the competence of a reviewer in judging the quality of a submission.

### 5.2 Genuinely present citations

In this work, we aim at decoupling citation bias from a genuine change in the scientific merit of a submission due to additional citation. For this, we account for the genuinely missing citations confounding factor C1 that manifests in reviewers *genuinely decreasing* their scores when their relevant past paper is not cited in the submission.

In principle, we could also consider a symmetric *genuinely present citations* confounding factor that manifests in reviewers *genuinely increasing* their scores when their relevant past work is adequately incorporated in the submission. However, while symmetric, these two confounding factors are different in an important aspect. When citation of a relevant work is missing from the submission, an author of that relevant work is in a better position to identify this issue than other reviewers and this asymmetry of information can bias the analysis. However, when citation of a relevant work is present in the paper, all reviewers observe this signal as they read the paper. The presence of the shared source of information reduces the aforementioned asymmetry across reviewers and alleviates the corresponding bias.

With this motivation, in this work we do not specifically account for the genuinely present citations confounding factor, but we urge the reader to be aware of our choice when interpreting the results of our study.

## 5.3 Fidelity of citation relationship

Our analysis pertains to citation relationships between the submitted papers and the reviewers. In order to ensure that reviewers who are cited in the submissions are identified correctly, we developed a custom parsing tool. Our tool uses PDF text mining to (i) extract authors of papers cited in a submission (all common citation formats are accommodated) and (ii) match these authors against members of the reviewer pool. We note that there are several potential caveats associated with this procedure which we now discuss:

- **False positives**. First, reviewers' names are not unique identifiers. Hence, if the name of a reviewer is present in the reference list of a submission, we cannot guarantee that it is the specific ICML or EC reviewer cited in the submission. To reduce the number of false positives, we took the following approach. First, for each reviewer we defined a *key*:

    {LAST NAME}_{FIRST LETTER OF FIRST NAME}

    Second, we considered all reviewers whose *key* is not unique in the conference they review for. For these reviewers, we manually verified all assigned (submission, reviewer) pairs in which reviewers were found to be CITED by our automated mechanism. We found that about 50% of more than 250 such cases were false positives and corrected these mistakes, ensuring that the analysis data did not have false positives among reviewers with non-unique values of their *key*.
    Third, for the remaining reviewers (those whose *key* was unique in the reviewer pool), we sampled 50 (submission, CITED reviewer) pairs from the actual assignment and manually verified the citation relationship. Among 50 target pairs, we identified only 1 false positive case and arrived at the estimate of 2% of false positives in our analysis.

- **False negatives**. In addition to false positives, we could fail to identify some of the CITED reviewers. To estimate the fraction of false negatives, we sampled 50 (submission, UNCITED reviewer) pairs from the actual assignment and manually verified the citation relationship. Among these 50 pairs we did not find any false negative case, which suggests that the number of false negatives is very small.

Finally, we note that both false positives and false negatives affect the power, but not the false alarm probability of our analysis. Thus, the conclusions of our analysis are stable with respect to imperfections of the procedure used to establish the citation relationship.

## 5.4 Generalizability of the results

As discussed in Section 3, in this experiment we used submissions that were assigned to at least one CITED and one UNCITED reviewers and satisfied other inclusion criteria (see Data Filtering in Section 3.2.2). We now perform some additional analysis to juxtapose the population of submissions involved in our analysis to the general population of submissions.

Fig 1 compares distributions of mean overall scores given in initial reviews between submissions that satisfied the inclusion criteria of our analysis and submissions that were excluded from consideration. First, observe that Fig 1a suggests that in terms of the overall scores, ICML submissions used in the analysis are representative of the general ICML submission pool. However, in EC (Fig 1b), the submissions that were used in the analysis received on average higher scores than those that were excluded. Thus, we urge the reader to keep in mind that our analysis of the EC data may not be applicable to submissions that received lower scores.

One potential reason of the difference in generalizability of our EC and ICML analyses is the intervention we took in EC to increase the sample size. Indeed, by maximizing the number

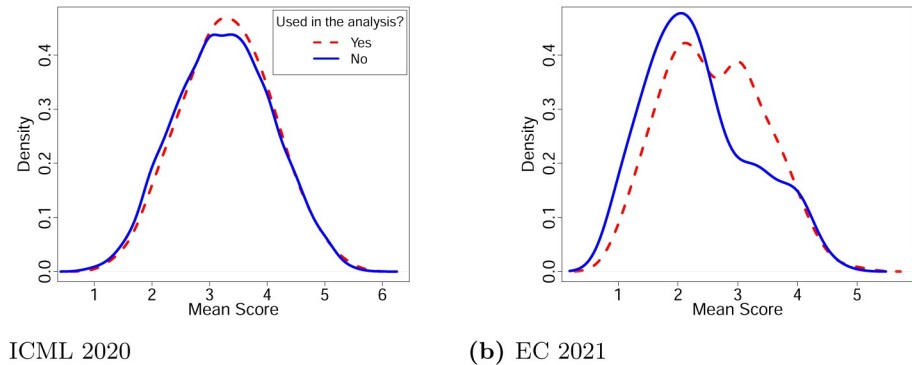

**(a)** ICML 2020                    **(b)** EC 2021

**Fig 1. Distribution of mean overall scores given in initial reviews with a breakdown by whether a submission is used in our analysis or not.**

of submissions that are assigned to at least one CITED reviewer we could include most of the submissions that are *relevant* to the venue in the analysis, which results in the observed difference in Fig 1b.

## 5.5 Spurious correlations induced by reviewer identity

In peer review, each reviewer is assigned to several papers. Our analysis implicitly assumes that conditioned on `quality`, `expertise`, `preference`, `seniority` characteristics, and on the value of the `citation` indicator, evaluations of different submissions made by the same reviewer are independent. Strictly speaking, this assumption may be violated by correlations introduced by various characteristics of the reviewer identity (e.g., some reviewers may be lenient while others are harsh). To fully alleviate this concern, we would need to significantly reduce the sample size by requiring that each reviewer contributes to at most one (submission, reviewer) pair used in the analysis. Given otherwise limited sample size, this requirement would put a significant strain on our testing procedure. Thus, in this work we follow previous empirical studies of the peer-review procedure [17, 28, 32] and tolerate such potential spurious correlations. We note that simulations performed by [31] demonstrate that unless reviewers contribute to dozens of data points, the impact of such spurious correlations is limited. In our analysis, reviewers on average contributed to 1.8 (submission, reviewer) pairs in ICML, and to 5.5 (submission, reviewer) pairs in EC, thereby limiting the impact of this caveat.

## 5.6 Counteracting the effect

Our analysis raises an open question of counteracting the effect of citation bias in peer review. For example, one way to account for the bias is to increase the awareness about the bias among members of the program committee and add citation indicators to the list of information available to decision-makers. Another option is to try to equalize the number of CITED reviewers assigned to submissions. Given that [24] found citation indicator to be a good proxy towards the quality of the review, enforcing the balance across submissions could be beneficial for the overall fairness of the process. More work may be needed to find more principled solutions against citation bias in peer review.

## Supporting information

**S1 Appendix. Controlling for confounding factors.**
(PDF)

**S2 Appendix. Details of the parametric inference.**
(PDF)

**S3 Appendix. Details of the non-parametric inference.**
(PDF)

**S4 Appendix. Model diagnostics.**
(PDF)

**S1 File.**
(PDF)

## Acknowledgments

We appreciate the efforts of all reviewers involved in the review process of ICML 2020 and EC 2021. We thank Valerie Ventura for useful comments on the design of our analysis procedure.

## Author Contributions

**Conceptualization:** Ivan Stelmakh, Charvi Rastogi, Ryan Liu, Shuchi Chawla, Federico Echenique, Nihar B. Shah.

**Data curation:** Ivan Stelmakh, Charvi Rastogi, Ryan Liu, Shuchi Chawla, Federico Echenique.

**Formal analysis:** Ivan Stelmakh, Charvi Rastogi.

**Funding acquisition:** Nihar B. Shah.

**Investigation:** Ivan Stelmakh, Charvi Rastogi, Ryan Liu, Nihar B. Shah.

**Methodology:** Ivan Stelmakh, Charvi Rastogi, Ryan Liu, Nihar B. Shah.

**Project administration:** Shuchi Chawla, Federico Echenique, Nihar B. Shah.

**Supervision:** Shuchi Chawla, Federico Echenique, Nihar B. Shah.

**Visualization:** Ivan Stelmakh, Charvi Rastogi.

**Writing – original draft:** Ivan Stelmakh, Charvi Rastogi, Ryan Liu, Nihar B. Shah.

**Writing – review & editing:** Ivan Stelmakh, Charvi Rastogi, Ryan Liu, Shuchi Chawla, Federico Echenique, Nihar B. Shah.

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
