## [Decision Letter · Decision Letter 0]

24 Jan 2023

PONE-D-22-31312Cite-seeing and Reviewing: A Study on Citation Bias in Peer ReviewPLOS ONE

Dear Dr. Rastogi,

Thank you for submitting your manuscript to PLOS ONE. After careful consideration, we feel that it has merit but does not fully meet PLOS ONE’s publication criteria as it currently stands. Therefore, we invite you to submit a revised version of the manuscript that addresses the points raised during the review process. Congratulations on an excellent submission. While I have selected 'minor revisions,' this is solely so you might have a chance to respond to the two very minor comments by reviewer 2. I think these are good questions that would be very easy to address - really changing a word here or there. In addition, I would ask for a sentence or two that is more clarifying on how you're testing causality. I understand removing possible confounders, but I would still be a little more guarded in describing this as concluding causation. However, all these revisions are entirely up to you and your team. Once it is resubmitted, I will be very happy to accept the paper without sending it back out for review.

We look forward to receiving your revised manuscript.

Kind regards,

Lorien Shana Jasny

Academic Editor

PLOS ONE

Journal Requirements:

“This work was supported by NSF CAREER award 1942124. Charvi Rastogi was partially supported by a J.P. Morgan AI research fellowship.

NSF CAREER award 1942124 was awarded to Nihar Shah (https://www.nsf.gov/awardsearch/showAward?AWD_ID=1942124&HistoricalAwards=f alse)

J.P. Morgan AI research fellowship was awarded to Charvi Rastogi (https://www.jpmorgan.com/technology/artificial-intelligence/research-awards/phd-fellowship-2021)

The funders had no role in study design, data collection and analysis, decision to publish, or preparation of the manus”

“We appreciate the e↵orts of all reviewers involved in the review process of ICML 2020 and EC 2021. We thank Valerie Ventura for useful comments on the design of our analysis procedure. The experiment was reviewed and approved by an Institutional Review Board. This work was supported by NSF CAREER award 1942124. CR was partially supported by a J.P. Morgan AI research fellowship”

“his work was supported by NSF CAREER award 1942124. Charvi Rastogi was partially supported by a J.P. Morgan AI research fellowship.

NSF CAREER award 1942124 was awarded to Nihar Shah (https://www.nsf.gov/awardsearch/showAward?AWD_ID=1942124&HistoricalAwards=f alse)

J.P. Morgan AI research fellowship was awarded to Charvi Rastogi (https://www.jpmorgan.com/technology/artificial-intelligence/research-awards/phd-fellowship-2021)”

Additional Editor Comments (if provided):

Reviewers' comments:

Reviewer's Responses to Questions

**Comments to the Author**

1. Is the manuscript technically sound, and do the data support the conclusions?

Reviewer #1: Yes

Reviewer #2: Yes

2. Has the statistical analysis been performed appropriately and rigorously? 

Reviewer #1: Yes

Reviewer #2: Yes

3. Have the authors made all data underlying the findings in their manuscript fully available?

Reviewer #1: Yes

Reviewer #2: No

4. Is the manuscript presented in an intelligible fashion and written in standard English?

Reviewer #1: Yes

Reviewer #2: Yes

5. Review Comments to the Author

Reviewer #1: Thanks for a nice paper that adds to the literature on peer review and if/how biased the review process is. For me the write-up is very clear and appropriate, and as I appreciate reviewers who do not write *useless* suggestions for improvements on a paper that needs none, I also refrain from that.

Reviewer #2: In this well-written manuscript, Stelmakh et al. described an interesting experiment on citation bias in the peer review. They demonstrated that all else being equal, reviewers are more likely to give better evaluations for submissions that cite their work than those that do not. The research is carefully done, and the conclusion has profound implications for scientific evaluation. Therefore, I recommend acceptance without further delay. I do also recommend some minor modifications of the text, listed below:

1. The paper starts with the possibility that authors might intentionally cite potential reviewers' work to get their papers accepted more easily. However, in most cases, authors can't know and have little power to choose reviewers. Therefore, the current paper's results reflect the reviewers' behaviors more than the authors'. I recommend framing the bias as the bias of reviewers. But I do agree that under some circumstances, this paper's results have implications for authors' behavior, too—in many reviewing systems, the identity of editors is known to the authors. Authors can add this to their discussion;

2.Authors use the term 'quality' to describe the confounding factor 2. However, from the description, confounding factor 2 is more about relevance with respect to the specific conferences instead of quality. Specifically, novel, high-quality work might sometimes seem out of the scope of a specific venue and hence be rejected. I recommend using the term 'relevance' or 'relative quality' to describe the confounding factor 2 and add some elaboration in the corresponding section;

Overall, this article is excellent and worth a speedy publication.

6. PLOS authors have the option to publish the peer review history of their article (what does this mean?). If published, this will include your full peer review and any attached files.

Reviewer #1: No

Reviewer #2: No

---

## [Author Response · Author response to Decision Letter 0]

10 Mar 2023

Editor: 

Thank you for your positive feedback! We appreciate your consideration of our work. 

Regarding causal conclusion: We mention the caveats involved in our causal conclusion in the Discussion section under the paragraph heading: “Observation Caveat”. We reproduce key points made in that paragraph.

“First, we want to underscore that, while we try to carefully account for various confounding factors and our analysis employs different techniques, our study remains observational. Thus, the usual caveat of unaccounted confounding factors applies to our work. The main assumption that we implicitly make in this work is that the list of confounding factors C1–C5 is (i) exclusive and (ii) can be adequately modeled with the variables we have access to. “

Reviewer 1: 

Thank you for your positive feedback, and your time and consideration in reviewing our submission! 

Reviewer 2:

Thank you for your positive feedback, and for your suggestions to improve the submission. We incorporate your suggestions and provide a response for each, below: 

1. Framing of bias as bias of reviewers: Thank you for pointing this out. We modified and reorganized the introduction section to clarify our framing. The edited text is in blue in the revised and marked up version. 

2. Term for quality: As the reviewer rightly points out that the confounding factor termed “quality” in the manuscript refers to the relevant quality of the paper for the corresponding venue. We make this correction in our manuscript (quality → relevant quality) and add an explanation, reproduced here: 

“Quality: Relevant quality of the submission for the publication venue considered. We note that this quantity can be different from the quality of the submission independent of the publication venue.”

---

## [Editor Report · Decision Letter 1]

21 Mar 2023

Cite-seeing and Reviewing: A Study on Citation Bias in Peer Review

PONE-D-22-31312R1

Dear Dr. Rastogi,

We’re pleased to inform you that your manuscript has been judged scientifically suitable for publication and will be formally accepted for publication once it meets all outstanding technical requirements.

Kind regards,

Lorien Shana Jasny

Academic Editor

PLOS ONE
---

## [Editor Report · Acceptance letter]

26 Jun 2023

PONE-D-22-31312R1 

Cite-seeing and Reviewing: A Study on Citation Bias in Peer Review 

Dear Dr. Stelmakh:

I'm pleased to inform you that your manuscript has been deemed suitable for publication in PLOS ONE. Congratulations! Your manuscript is now with our production department. 

Kind regards, 

on behalf of

Dr. Lorien Shana Jasny 

Academic Editor

PLOS ONE